# Biventricular Myocardial Strain Analysis in Patients with Pulmonary Arterial Hypertension Using Cardiac Magnetic Resonance Tissue-Tracking Technology

**DOI:** 10.3390/jcm11082230

**Published:** 2022-04-15

**Authors:** Jibin Cao, Simiao Li, Lingling Cui, Kexin Zhu, Huaibi Huo, Ting Liu

**Affiliations:** Department of Radiology, The First Hospital of China Medical University, 155 Nanjing Bei Street, Heping District, Shenyang 110001, China; caojb@cmu1h.com (J.C.); lsm957545773@163.com (S.L.); cuill@cmu1h.com (L.C.); springzhuwork@163.com (K.Z.)

**Keywords:** cardiac magnetic resonance tissue tracking, CMR-TT, pulmonary arterial hypertension, PAH

## Abstract

To evaluate both left and right ventricular (LV and RV) function in patients with pulmonary arterial hypertension (PAH) using cardiac magnetic resonance tissue-tracking (CMR-TT) technology and explore its clinical value. Methods: A total of 79 participants (including 47 patients with PAH and 32 healthy controls) underwent cardiac magnetic resonance imaging (CMRI) with a short-axis balanced steady-state free precession (SSFP) sequence. The biventricular cardiac function parameters and strain parameters were obtained by postprocessing with CVI42 software. A comparative analysis was performed between the LV and RV strain parameters in all PAH patients and in PAH patients with reduced or preserved cardiac function. Results: The results showed preferable repeatability of CMR-TT in analyzing the global radial strain (GRS), circumferential strain (GCS), and longitudinal strain (GLS) of the left and right ventricles in the PAH group. The GRS, GCS, and GLS of the left and right ventricles except for LV GRS (LVGRS) of PAH patients were significantly lower than those of healthy controls (*p* < 0.05 for all). The GRS and GCS of the left and right ventricles showed a moderate correlation in the PAH group (r = 0.323, *p* = 0.02; r = 0.301, *p* = 0.04, respectively). PAH patients with preserved RV function (*n* = 9) showed significantly decreased global and segmental RS, CS, and LS of the right ventricles than healthy controls (*p* < 0.05 for all), except for basal RVGCS (RVGCS-b, *p* = 0.996). Only the LVGLS was significantly different between the PAH patients with preserved LV function (*n* = 32) and the healthy controls (−14.23 ± 3.01% vs. −16.79 ± 2.86%, *p* < 0.01). Conclusions: As a nonradioactive and noninvasive technique, CMR-TT has preferable feasibility and repeatability in quantitatively evaluating LV and RV strain parameters in PAH patients and can be used to effectively detect early biventricular myocardial damage in patients with PAH.

## 1. Introduction

Pulmonary arterial hypertension (PAH) is defined by a mean pulmonary artery pressure (mPAP) ≥ 25 mmHg at rest associated with pulmonary artery wedge pressure (PAWP) ≤ 15 mmHg, leading to right ventricular (RV) dysfunction and a rising afterload [1,2]. RV function failure is the main cause of death in PAH patients, and there is increasing evidence of the importance of right ventricular systolic parameters [3,4]. It has been reported that RV function is one of the major prognostic predictors of PAH [5,6,7].

The gold standard for diagnosing PAH is right heart catheterization (RHC) [8], which can be used to evaluate pulmonary hemodynamics with good repeatability [9]. Considering that RV function is the main determinant of the clinical outcome and that RHC is invasive, a noninvasive imaging method is needed to evaluate RV function in clinical practice. Most related studies have focused on RV function; it is well known that increased RV pressure can lead to changes in interventricular septum morphology [10] and that the morphology and function of the left and right ventricles are mutually affected. Therefore, the noninvasive evaluation of LV and RV function is vital.

Echocardiography and cardiac magnetic resonance imaging (CMRI) are common noninvasive methods for cardiac examination [11]. CMRI has a higher tissue resolution than echocardiography and is the gold standard for measuring heart volume and evaluating cardiac function [12,13]. However, the image quality is greatly affected by the patient’s respiration and heart rate, the examination time is longer, and the price is higher. In recent years, myocardial strain parameters have been obtained by tissue-tracking (TT) technology by postprocessing conventional CMRI data, and this method does not require additional sequences or images [14]. Therefore, it has been widely used in a variety of heart diseases with strong repeatability [15,16]. Strain parameters can reflect global and local myocardial deformation. A recent study has shown that there is heterogeneity in local RV function in patients with pulmonary hypertension (PH) before a decrease in the RV ejection fraction (RVEF) is observed, so evaluating RV myocardial deformation can reveal subclinical cardiac dysfunction [14]. Studies have also found that myocardial strain parameters are related to the prognosis of patients with PH [14,17].

At present, studies on ventricular strain in patients with PAH using CMRI have mostly focused on a single ventricle (particularly the right ventricle), while fewer studies have focused on biventricular myocardial strain analysis. The purpose of this study was to retrospectively analyze the correlation between the LV and RV strain, as well as the ventricular strain changes associated with cardiac function preservation among PAH patients.

## 2. Materials and Methods

### 2.1. Participants

We retrospectively analyzed 47 patients diagnosed with PAH (World Health Organization (WHO) Group 1) who underwent CMRI between February 2017 and March 2020. The inclusion criteria were those established according to the 2015 ESC/ERS guidelines [8]. All patients enrolled in this study were diagnosed with PAH by echocardiography (ECG), history taking, and RHC. Thirty-nine PAH subjects had idiopathic PAH, six had PAH associated with systemic lupus erythematosus (SLE), and two had PAH associated with systemic sclerosis. In our cohort, PAH was defined as mPAP ≥25 mmHg at rest and PAWP was ≤15 mmHg obtained from RHC. The exclusion criteria were as follows: (1) any underlying cardiomyopathy (such as hypertrophic, amyloid, or ischemic cardiomyopathy); (2) arrhythmia; (3) lung disease/hypoxia, chronic thromboembolism; (4) massive pericardial effusion; (5) substandard image quality; and (6) contraindication to MRI. As controls, 32 healthy subjects without any cardiovascular disease symptoms and with normal ECG results were included.

### 2.2. MRI Acquisition

CMRI was performed using a 1.5-T system (Signa EXCITE HDx; GE Health care, Milwaukee, MI, USA) and a 3.0-T MR system (MAGNETOM Verio, Siemens Health care, Erlangen, Germany) with a 32-channel phased-array surface coil and MR-compatible ECG and respiratory gating. Multiphase cine imaging was performed using a standard steady-state free precession (SSFP) pulse sequence in the LV two-chamber and four-chamber long-axis views and in the biventricular short-axis views with breath holding (field of view (FOV) = 340 × 360 mm^2^, matrix = 256 × 192, TR = 51.45 ms, TE = 1.51 ms; short-axis planes: 10 or 8 mm thickness, 0 mm gap, matrix = 216 × 256) (Figure 1).

### 2.3. Image Postprocessing

#### 2.3.1. Cardiac Function Parameters

The acquired MRI data were processed using CVI42 (version 5.3.4, Circle Cardiovascular Imaging, Calgary, AB, Canada) by 2 experienced radiologists with more than 3 years of CMRI experience. The endometrium and epicardium of the left and right ventricles were semiautomatically delineated on the short axis of the ventricles at the end of diastole and end of systole, and the parameters of the cardiac function of both ventricles were obtained after making appropriate adjustments. The cardiac functional parameters included the end-diastolic volume (EDV), end-systolic volume (ESV), and EF.

#### 2.3.2. Myocardial Strain Parameters

Biventricular myocardial strain parameters were measured using the CVI42 TT technique. The investigators selected the diastolic phase of the left ventricle, semiautomatically outlined the endocardium of both ventricles, marked the septum in the short-axis view of both ventricles, marked the valve plane and apical position in the standard four-chamber view and the two-chamber view of the left ventricle, and then automatically generated the 2D strain parameters and strain curves of both ventricles, as shown in Figure 2.

### 2.4. Statistical Analysis

SPSS version 22.0 software (SPSS, Chicago, IL, USA) was used to conduct the statistical analyses. Continuous data were expressed as the mean ± standard deviation (SD). The Shapiro–Wilk test was used to test the normality of the strain parameters, and two independent sample *t* tests or nonparametric rank-sum tests were selected for comparisons between the two groups. We used one-way ANOVA for comparisons among PAH patients with a reduced EF, PAH patients with a preserved EF, and healthy controls. The relationship between LV and RV strain parameters in the PAH group was assessed using Pearson’s correlation analysis. *p* < 0.05 was considered to indicate a statistically significant difference.

### 2.5. Repeatability Test

Twenty patients with PAH were randomly selected, and CMR-TT postprocessing was performed by two radiologists who were blinded to the information of the patients. Consistency was tested by Bland–Altman analysis, along with an interclass correlation coefficient (ICC) to calculate the 95% limits of mean deviation and consistency.

## 3. Results

### 3.1. General Information

Among the 47 patients in the PAH group, there were 9 males and 38 females. In the healthy control group, there were 20 males and 12 females. The age and cardiac function parameters in the two groups are shown in Table 1.

### 3.2. Reproducibility Analysis

The global LV and RV strain were reproducible at the interobserver level. Bland–Altman Plots for global strain of both ventricles are shown in Figure 3. The 95% CI and ICC for strain parameters are summarized in Table 2. The radiologists showed good agreement regarding their interpretations. The results showed preferable repeatability of CMR-TT technology in analyzing the global radial strain (GRS), circumferential strain (GCS), and longitudinal strain (GLS) of the left and right ventricles in the PAH group (Table 2), and the LVGCS showed the best interobserver agreement (95% CI, 0.956–0.994).

### 3.3. Strain Analysis in the PAH and Control Groups

According to the methods of echocardiographic studies, strain is a vector quantity that expresses myocardial deformation, with positive values indicating thickening and lengthening of the myocardium and negative values indicating thinning and shortening of the myocardium [18]. The GLS of both ventricles and the RVGRS and RVGCS of PAH patients were significantly lower than those of healthy controls (*p* < 0.001) (Table 3). We further found significant differences in LVGCS between PAH patients and healthy controls (*p* = 0.027), but no significant difference in LVGRS was observed (*p* = 0.077) (Table 3).

### 3.4. Correlation Analysis of LV and RV Strain in the PAH Group

The GRS and the GCS of the left and right ventricles showed a moderate correlation in the PAH group (r = 0.323, *p* = 0.02; r = 0.301, *p* = 0.04, respectively). The GLS of the left and right ventricles was not statistically correlated (r = −0.138, *p* = 0.354) (Figure 4).

### 3.5. RV Myocardial Strain Analysis in PAH Patients with Preserved and Reduced RV Function

The RV strain parameters in the three groups are listed in Table 4. The RVEF < 40% group had a significantly lower absolute RVGCS (−7.71 ± 2.90%), apical RVGCS (RVGCS-a, −6.38 ± 2.62%), and mid-segment RVGCS (RVGCS-m, −8.05 ± 4.40%) than both the RVEF ≥ 40% group and the healthy control group (*p* < 0.05 for all). Compared to that of control subjects, RVGRS, RVGLS, the basal, mid-segment, and apical of RVGRS (RVGRS-b, RVGRS-m, RVGRS-a) of PAH patients with preserved RV function was significantly impaired (12.31 ± 4.80% vs. 24.33 ± 7.98%; −11.13 ± 5.62% vs. −20.08 ± 6.41%; 12.37 ± 4.28% vs. 18.48 ± 7.53%; 13.26 ± 6.32% vs. 27.99 ± 11.90% and 14.99 ± 4.94% vs. 38.07 ± 18.69%, respectively, *p* < 0.05 for all), while that of PAH patients with reduced RV function showed no significant difference. There were no significant differences in the basal RVGCS (RVGCS-b) among the three groups (*p* = 0.996) (Table 4).

### 3.6. LV Myocardial Strain Analysis in PAH Patients with Preserved and Reduced LV Function

Among the 47 patients in the PAH group, there were 32 patients with preserved LV function (LVEF ≥ 60%). All LV strain parameters in PAH patients with reduced LV function were significantly lower than those in both PAH patients with preserved LV function and in healthy controls (*p* < 0.01 for all). There was a significant difference in the LVGLS between the PAH patients with preserved LV function and the healthy controls (−14.23 ± 3.01% vs. −16.79 ± 2.86%, *p* < 0.01) (Table 5).

## 4. Discussion

### 4.1. Feasibility and Repeatability of CMR-TT

Our findings provide data supporting the role of quantitative tissue techniques in noncontrast methods for identifying and monitoring patients with PAH. The strain parameters of the PAH patients in this study were measured independently by two investigators for the same 20 randomly selected patients. The obtained GRS, GCS, and GLS of the left and right ventricles showed good correlations in the Bland–Altman analysis, along with an interclass correlation coefficient (ICC) indicating that the CMR-TT technique is feasible and reproducible for the evaluation of biventricular myocardial strain parameters in patients with PAH [19,20].

### 4.2. Biventricular Myocardial Strain Analysis in PAH Patients

Currently, several studies on RV strain in patients with PAH can be found. de Siqueira et al. [14] found significant differences in the RVGCS and RVGLS of PH Group 1 and Group 5 patients compared with controls using characteristic tracing analysis, but the etiologies of PH included in the study were heterogeneous. A study by D’andrea et al. [21] revealed that RVGLS was an important prognostic factor. Haeck et al. [17] evaluated 150 patients with PH compared with patients with a higher GLS and demonstrated that RVGLS was a triple determinant of all-cause mortality risk (threshold < −19%).

Strain has been suggested as a potential metric for evaluating early or subclinical changes in LV function. It performs better than the LVEF in reflecting myocardial deformation and has already been incorporated into alternative measurements of therapeutic results [22,23]. Kallianos et al. [24] compared LV strain parameters between PH patients and controls and found a significant difference in LVGCS. In a long-term follow-up study [25], the LV volume per beat and LVGLS were correlated with RV dysfunction and poor clinical outcomes.

The analysis of biventricular GLS performed by Lindholm et al. [26] in PH patients with systemic sclerosis showed that the altered biventricular GLS was mainly caused by PH. In this study, we first analyzed the biventricular GRS, GCS, and GLS in PAH patients and found that strain parameters in both ventricles except for LVGRS were significantly different from those in the controls in all three directions. A similar study of biventricular strain in PAH performed by Lin et al. [27] found that RVGLS was significantly reduced in PAH (*p* = 0.02), and the close relation between RVESV index with LVGCS and LVGRS (r = −0.65 and r = −0.70, *p* < 0.05) provides evidence of systolic ventriculo-ventricular interaction in PAH. Another recent study performed by Kallianos et al. [24] revealed that among patients with PH, there was a statistically significant association between LVGCS and RV end-diastolic volume index (RVEDVI; r^2^ = 0.27, *p* = 0.047). Our findings are consistent with those of several previous studies, in that LVGCS and LVGRS but not GLS are significantly correlated with RVGCS and RVGRS, respectively (r = 0.323 and r = 0.301, *p* < 0.05). It suggested that the relationship between biventricular strain parameters might be related to inter-ventricular dependence. A potential explanation could be that the reverse septal curvature and a more D-shaped LV in patients with PAH might affect circumferential more than longitudinal myocardial fiber shortening [28], this then results in the myocardium’s radial thickening.

### 4.3. Strain Analysis of PAH Patients with Preserved RV Function

The RVGRS, RVGCS, and RVGLS in PAH patients with preserved RV function were significantly different from those in the controls. This result is also consistent with the study reported by de Siqueira [14], in which the RVGLS and RVGCS were found to be significantly different between PH patients in the RVEF > 50% group and the healthy controls. Our analysis of RV strain in PAH patients with reduced and preserved RV function compared to controls demonstrates the feasibility of using CMR-TT to detect RV dysfunction in patients with known PAH before the EF significantly declines.

### 4.4. Strain Analysis of PAH Patients with Preserved LV Function

Most related studies have focused on RV function in PAH patients, and there have been no definitive studies on strain analysis in PAH patients with preserved LV function. In this study, only the GLS was significantly different between PAH patients with preserved LV function and the controls, while the LVGRS and LVGCS showed no significant difference between these two groups. Kallianos et al. [24] reported on the CMR evaluation of LV myocardial strain in 16 PH patients including Group 1–5, and stated there were no significant differences in the LVEF between the PH and control groups (60.2 ± 11.0% and 61.9 ± 4.5%, respectively, uncorrected *p* = 0.634, age-corrected *p* = 0.150), while there was a significant difference in the GLS, which is similar to the results of our present study. In addition, Padervinskiene et al. [29] analyzed one-year survival in patients with precapillary PH and found a significant difference in LVGLS between the one-year death group and the one-year survival group. These results may indicate that LVGLS is associated with a poor prognosis. Above all, PAH patients also have impaired LV function, and our studies indicated that the decrease in myocardial strain could occur earlier than the decrease in the LVEF, especially in terms of LVGLS, in PAH patients with preserved LV function.

### 4.5. Limitations

This was a single-center, retrospective analysis of patients with PAH who underwent clinical CMRI. The number of PAH patients who underwent clinical CMRI during the study was restricted. This study’s other limitation might be that there were differences in the sex ratio between the control and patient groups. Previous studies have shown sex differences in the LV and RV strain in healthy individuals [30,31], thus differences in strain parameters between the two groups might be influenced by sex. Finally, only strain parameters were studied, and the strain rates of PAH patients should be further analyzed.

## 5. Conclusions

As a nonradioactive and noninvasive technique, CMR-TT has preferable feasibility and repeatability for quantitatively evaluating LV and RV strain parameters in PAH patients and can be used to effectively detect early biventricular myocardial damage in PAH patients.

## Figures and Tables

**Figure 1 jcm-11-02230-f001:**
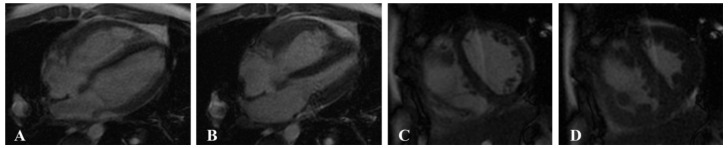
Multiphase cine imaging was performed using a standard steady-state free precession (SSFP) pulse sequence in a 14-year-old male patient with pulmonary arterial hypertension (PAH). Four-chamber long-axis view at the end of diastole (**A**) and end of systole (**B**) and biventricular short-axis views at the end of the diastole (**C**) and the end of the systole (**D**).

**Figure 2 jcm-11-02230-f002:**
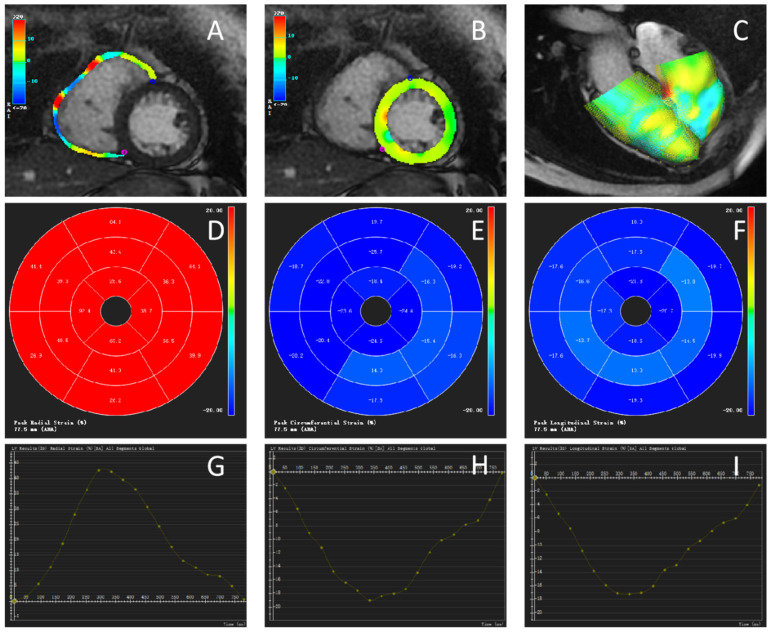
Cardiac magnetic resonance tissue tracking (CMR-TT) in the same patient with PAH shown in Figure 1. (**A**–**C**) Global radial strain of the ventricles in diastole. Color coding represents the tracking for the ventricle. (**A**) Right ventricle. (**B**) Left ventricle. (**C**) 3D model of the left and right ventricles in diastole. The color coding represents the epicardial contours. (**D**–**F**) Polar map of 2002 American Heart Association segmentation. (**D**) Radial strain (RS). (**E**) Circumferential strain (CS). (**F**) Longitudinal strain (LS). (**G**–**I**) Curve showing the strain values of the left ventricle in 25 phases, including the peak value. (**G**) RS. (**H**) CS. (**I**) LS.

**Figure 3 jcm-11-02230-f003:**
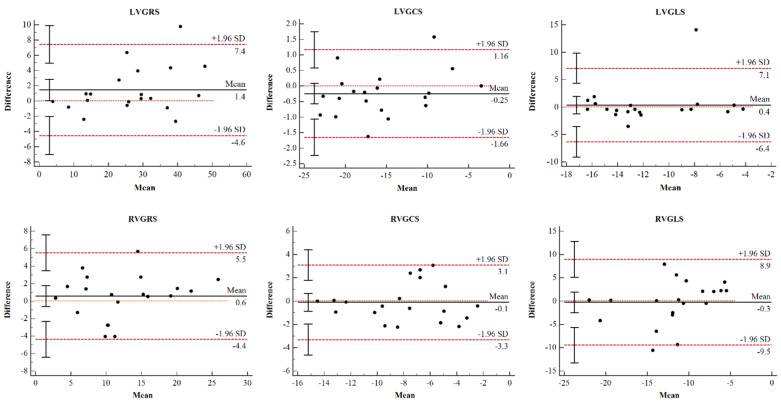
Bland-Altman plots for interobserver variability. Bland-Altman plots for interobserver variability obtained for global LV strain and global RV strain. LV, left ventricular; RV, right ventricular; GRS, global radial strain; GCS, global circumferential strain; GLS, global longitudinal strain.

**Figure 4 jcm-11-02230-f004:**
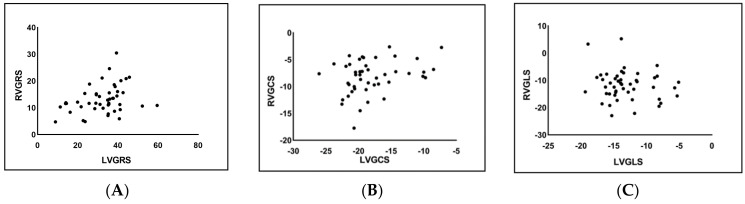
Scatter plots of global radial strain (GRS) (**A**), global circumferential strain (GCS) (**B**), and global longitudinal strain (GLS) (**C**) in the left and right ventricles of PAH patients.

**Table 1 jcm-11-02230-t001:** Baseline clinical characteristics and cardiac function parameters in the PAH and healthy control groups.

	PAH Group (*n* = 47)	Healthy Control Group (*n* = 32)	*p*
Age (years)	49.32 ± 14.23	53.13 ± 15.06	0.258
Gender (male/female)	9/38	17/15	<0.001
Height (cm)	164.00 ± 6.06	167.19 ± 7.28	0.059
Weight (kg)	60.26 ± 8.64	61.78 ± 10.27	0.167
BMI (kg/m^2^)	22.28 ± 1.69	21.94 ± 2.13	0.115
Heart rate (bpm)	69.68 ± 5.28	71.12 ± 5.92	0.561
LVEF %	62.29 ± 10.97	67.84 ± 6.90	0.013
LVEDV (mL)	85.60 ± 32.88	116.61 ± 24.70	<0.001
LVESV (mL)	34.31 ± 26.20	37.74 ± 12.04	0.492
RVEF %	29.80 ± 8.30	47.92 ± 7.04	<0.001
RVEDV (mL)	132.72 ± 43.41	106.16 ± 27.13	0.003
RVESV (mL)	94.74 ± 36.93	54.70 ± 16.53	<0.001

Note: PAH, pulmonary arterial hypertension; BMI, body mass index; LV, left ventricular; RV, right ventricular; EDV, end-diastolic volume; ESV, end-systolic volume; EF, ejection fraction.

**Table 2 jcm-11-02230-t002:** Interobserver consistency analyses.

	ICC	95% CI	*p*
LVGRS %	0.976	0.932–0.991	<0.01
LVGCS %	0.985	0.956–0.994	<0.01
LVGLS %	0.819	0.540–0.929	<0.01
RVGRS %	0.957	0.894–0.983	<0.01
RVGCS %	0.947	0.867–0.979	<0.01
RVGLS %	0.759	0.382–0.905	<0.01

Note: ICC, interclass correlation coefficient; LV, left ventricular; RV, right ventricular; GRS, global radial strain; GCS, global circumferential strain; GLS, global longitudinal strain.

**Table 3 jcm-11-02230-t003:** Comparison of myocardial strain values between the PAH and healthy control groups.

	PAH Group (n = 47)	Healthy Control Group (n = 32)	p
LVGRS %	32.37 ± 10.60	36.31 ± 7.92	0.077
LVGLS %	−13.04 ± 3.49	−16.79 ± 2.86	<0.001
LVGCS %	−18.01 ± 4.21	−19.91 ± 2.90	0.027
RVGRS %	13.18 ± 5.21	24.33 ± 7.98	<0.001
RVGLS %	−11.81 ± 5.46	−20.08 ± 6.41	<0.001
RVGCS %	−8.19 ± 3.10	−13.30 ± 3.61	<0.001

Note: LV, left ventricular; RV, right ventricular; GRS, global radial strain; GCS, global circumferential strain; GLS, global longitudinal strain.

**Table 4 jcm-11-02230-t004:** Comparison of RV strain parameters among PAH patients with reduced and preserved RV function and healthy controls.

	RVEF < 40% Group (*n* = 38)	RVEF ≥ 40% Group (*n* = 9)	Healthy Control Group (*n* = 32)	*p*
RVGRS %	12.31 ± 4.80 ^b^	16.85 ± 5.57 ^b^	24.33 ± 7.98	<0.001
RVGLS %	−11.13 ± 5.62 ^b^	−14.70 ± 3.68 ^b^	−20.08 ± 6.41	<0.001
RVGCS %	−7.71 ± 2.90 ^ab^	−10.22 ± 3.26 ^b^	−13.30 ± 3.61	<0.001
RVGCS-b %	−7.38 ± 3.85	−7.58 ± 6.33	−7.45 ± 8.52	0.996
RVGCS-m %	−8.05 ± 4.40 ^ab^	−11.49 ± 3.04 ^b^	−15.36 ± 3.06	<0.001
RVGCS-a %	−6.38 ± 2.62 ^ab^	−13.43 ± 4.18 ^b^	−19.08 ± 5.49	<0.001
RVGRS-b %	12.37 ± 4.28 ^b^	14.34 ± 8.22 ^b^	18.48 ± 7.53	0.001
RVGRS-m %	13.26 ± 6.32 ^b^	18.34 ± 5.73 ^b^	27.99 ± 11.90	<0.001
RVGRS-a %	14.99 ± 4.94 ^b^	23.20 ± 8.96 ^b^	38.07 ± 18.69	<0.001

Note: RV, right ventricular; GRS, global radial strain; GCS, global circumferential strain; GLS, global longitudinal strain; GRS-a, apical GRS; GRS-m, mid-segment GRS; GRS-b, basal GRS; GCS-a, apical GCS; GCS-m, mid-segment GCS; GCS-b, basal GCS. ^a^ *p* < 0.05 vs. RVEF ≥ 40% group. ^b^ *p* < 0.05 vs. controls.

**Table 5 jcm-11-02230-t005:** Comparison of LV strain parameters among PAH patients with reduced and preserved LV function and healthy controls.

	LVEF < 60% Group (*n* = 15)	LVEF ≥ 60% Group (*n* = 32)	Healthy Control Group (*n* = 32)	*p*
LVGRS %	21.97 ± 7.41 ^ab^	37.24 ± 8.06	36.31 ± 7.92	<0.001
LVGLS %	−10.51 ± 3.14 ^ab^	−14.23 ± 3.01 ^b^	−16.79 ± 2.86	<0.001
LVGCS %	−14.01 ± 3.76 ^ab^	−19.89 ± 2.92	−19.91 ± 2.90	<0.001
LVGCS-b %	−14.04 ± 3.46 ^ab^	−18.74 ± 2.74	−18.81 ± 2.75	<0.001
LVGCS-m %	−13.64 ± 3.63 ^ab^	−19.26 ± 3.53	−19.80 ± 3.06	<0.001
LVGCS-a %	−17.33 ± 5.42 ^ab^	−24.82 ± 2.96	−24.43 ± 4.29	<0.001
LVGRS-b %	22.10 ± 7.00 ^ab^	33.57 ± 7.24	33.51 ± 7.31	<0.001
LVGRS-m %	20.59 ± 6.85 ^ab^	34.84 ± 9.66	35.34 ± 9.04	<0.001
LVGRS-a %	29.77 ± 14.37 ^ab^	57.17 ± 15.06	53.64 ± 18.10	<0.001

Note: LV, left ventricular; GRS, global radial strain; GCS, global circumferential strain; GLS, global longitudinal strain; GRS-a, apical GRS; GRS-m, mid-segment GRS; GRS-b, basal GRS; GCS-a, apical GCS; GCS-m, mid-segment GCS; GCS-b, basal GCS. ^a^ *p* < 0.01 vs. LVEF ≥ 60% group. ^b^ *p* < 0.01 vs. controls.

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
