# Peer review of "Biventricular Myocardial Strain Analysis in Patients with Pulmonary Arterial Hypertension Using Cardiac Magnetic Resonance Tissue-Tracking Technology"

_jcm, 2022, doi:10.3390/jcm11082230_

Round 1
Reviewer 1 Report
I think there may a typo in Abstract section: The GRS, GCS and GLS of the left and right ventricles except for LVGLS of PAH patients were significantly lower than those of healthy controls (P < 0.05 for all).
probably LVGLS may need to be replaced with LVGRS. please double check.
The following lines copied from manuscript: We further found significant differences in LVGCS between PAH patients and healthy controls (P = 0.027), but no significant difference in LVGRS was observed (P = 0.077) (Table 3).
Author Response
Thank you for the valuable suggestions. We have corrected the Abstract and Discussion as below:
Abstract:
The GRS, GCS and GLS of the left and right ventricles except for LVGRS of PAH patients were significantly lower than those of healthy controls (P < 0.05 for all).
Discussion :
4.2. Biventricular myocardial strain analysis in PAH patients
The analysis of biventricular GLS performed by Lindholm et al. [27] in PH patients with systemic sclerosis showed that the altered biventricular GLS was mainly caused by PH. In this study, we first analysed the biventricular GRS, GCS, and GLS in PAH patients and found that strain parameters in both ventricles except for LVGRS were significantly different from those in the controls in all three directions. A similar study of biventricular strain in PAH performed by Lin et al. [28] found that RVGLS was significantly reduced in PAH (P = 0.02), and the close relation between RVESV index with LVGCS and LVGRS (r = -0.65 and r = -0.70, P < 0.05) provides evidence of systolic ventriculo-ventricular interaction in PAH. Our findings are consistent with those of several previous studies.

Reviewer 2 Report
The authors investigated the feasibility and reproducibility of biventricular strain analyses in patients with PAH. They found that CMR tissue tracking had preferable feasibility and repeatability in quantitatively evaluating biventricular strain parameters in patients with PAH. Several concerns have been raised.
- For Table 1, a little bit more data would be required to show baseline characteristics.
- It might be unclear where the results of Bland-Altman test are displayed.
- How can we interpret the findings of Figure 3: correlation between right and left ventricles in strain data?
Author Response
- For Table 1, a little bit more data would be required to show baseline characteristics.
Thank you for the valuable suggestions. We have revised Table 1 as the reviewer’s sugguestion as below.
Table 1. Baseline clinical characteristics and cardiac function parameters in the PAH and healthy control groups.
|
|
PAH group (n = 47) |
Healthy control group (n = 32) |
P |
|
Age (years) |
49.32 ± 14.23 |
53.13 ± 15.06 |
0.258 |
|
Gender (male/female) |
9/38 |
17/15 |
< 0.001 |
|
Height (cm) |
164.00 ± 6.06 |
167.19 ± 7.28 |
0.059 |
|
Weight (kg) |
60.26 ± 8.64 |
61.78 ± 10.27 |
0.167 |
|
BMI (kg/m2) |
22.28 ± 1.69 |
21.94 ± 2.13 |
0.115 |
|
Heart rate (bpm) |
69.68 ± 5.28 |
71.12 ± 5.92 |
0.561 |
|
LVEF % |
62.29 ± 10.97 |
67.84 ± 6.90 |
0.013 |
|
LVEDV (ml) |
85.60 ± 32.88 |
116.61 ± 24.70 |
< 0.001 |
|
LVESV (ml) |
34.31 ± 26.20 |
37.74 ± 12.04 |
0.492 |
|
RVEF % |
29.80 ± 8.30 |
47.92 ± 7.04 |
< 0.001 |
|
RVEDV (ml) |
132.72 ± 43.41 |
106.16 ± 27.13 |
0.003 |
|
RVESV (ml) |
94.74 ± 36.93 |
54.70 ± 16.53 |
< 0.001 |
Note: LV, left ventricular; RV, right ventricular; EDV, end-diastolic volume; ESV, end-systolic volume; EF, ejection fraction.
- It might be unclear where the results of Bland-Altman test are displayed.
Both ICC and Bland–Altman analysis were included in our study.
We have edited the details about the Reproducibility analysis (Figure3) regarding the comments and change as below:
3.2. Reproducibility analysis
Global LV and RV strain were reproducible at interobserver level. Bland–Altman Plots for global strain of both ventricles are shown in Figure 3. The 95% CI and ICC for strain parameters are summarized in Table 2. The radiologists showed good agreement regarding their interpretations. The results showed preferable repeatability of CMR-TT technology in analysing the global radial strain (GRS), circumferential strain (GCS) and longitudinal strain (GLS) of the left and right ventricles in the PAH group (Table 2), and the LVGCS showed the best interobserver agreement (95% CI, 0.956-0.994).
Figure 3. Bland–Altman plots for interobserver variability. Bland–Altman plots for interobserver variability obtained for global LV strain and global RV strain. LV, left ventricular; right ventricular, RV; GRS, global radial strain; GCS, global circumferential strain; GLS, global longitudinal strain.
- How can we interpret the findings of Figure 3: correlation between right and left ventricles in strain data?
Thank you for the valuable comments. The discussion was revised based on the reviewer’s suggestion.
The analysis of biventricular GLS performed by Lindholm et al. [27] in PH patients with systemic sclerosis showed that the altered biventricular GLS was mainly caused by PH. In this study, we first analysed the biventricular GRS, GCS, and GLS in PAH patients and found that strain parameters in both ventricles except for LVGRS were significantly different from those in the controls in all three directions. A similar study of biventricular strain in PAH performed by Lin et al. [28] found that RVGLS was significantly reduced in PAH (P = 0.02), and the close relation between RVESV index with LVGCS and LVGRS (r = -0.65 and r = -0.70, P < 0.05) provides evidence of systolic ventriculo-ventricular interaction in PAH. Another recent study performed by Kallianos et al. [25] revealed that among patients with PH, there was a statistically significant association between LVGCS and RV end-diastolic volume index (RVEDVI; r2 = 0.27, P = 0.047). Our findings are consistent with those of several previous studies, in that LVGCS and LVGRS but not GLS are significantly correlated with RVGCS and RVGRS respectively (r = 0.323 and r = 0.301, P < 0.05). It suggested that the relationship between biventricular strain parameters might be related to inter-ventricular dependence. A potential explanation could be that the reverse septal curvature and a more D-shaped LV in patients with PAH might affect circumferential more than longitudinal myocardial fiber shortening [29], this then results in the myocardium’s radial thickening.

Round 2
Reviewer 2 Report
There are no further comments that should be addressed.
This manuscript is a resubmission of an earlier submission. The following is a list of the peer review reports and author responses from that submission.
Round 1
Reviewer 1 Report
Major comments:
Opening line of manuscript (Line number 36 of manuscript) says pulmonary hypertension is defined as mean PA >25 mmHg. This is wrong: During the 6th World Symposium on Pulmonary Hypertension, in France, Pulmonary hypertension definition was revised by lowering the threshold from ≥25 mmHg to >20 mmHg. It needs to be changed.
The major issue I have is that authors have chosen echo parameter for diagnosing pulmonary hypertension. In many patients RHC is needed to confirm pulmonary hypertension. Did authors confirmed diagnosis of PH with RHC? I saw that towards the end authors have mentioned it in the limitation which is a major limitation of the study.
It was good to know that strain patterns were different between patients with PH compared to healthy subjects. Can authors also compare strain results among PH patients with preserved RV function vs declined RV function? I will be interested in knowing these findings.
Minor comments:
Please review the spelling and grammar thoroghly throughout the paper. For example. I think in abstract, result section: line number 22, authors probably wanted to say left "ventricle" and not left "eye".
Line number 189 "Among the 65 patients in the PH group, there were 39 patients with LV function 188 preservation (RVEF ≥ 60%)". Authors probably want to say LVEF >60% in parenthesis.
Did authors look into whether worse GRS, GCS or GLS correlate with mortality? I understand this many not be the primary focus of study and may not have been collected.
Authors should highlight that it is a retrospective study and is subject to limitations of retrospective study and genralization of these results should be made with caution.
Author Response
Dear Editors and Reviewers,
We would like to thank you for your careful reading, helpful comments, and constructive suggestions, which has significantly improved the presentation of our manuscript. It's an honor to have chance to publish our work in Journal of Clinical Medicine. We have carefully considered all comments from the reviewers and revised our manuscript accordingly. The manuscript has also been double-checked, and the typos and grammar errors we found have been corrected.
The followings are our RESPONSE to DEPUTY EDITOR & REVIEWER COMMENTS.
Reviewer A
Major comments:
- Opening line of manuscript (Line number 36 of manuscript) says pulmonary hypertension is defined as mean PA >25 mmHg. This is wrong: During the 6th World Symposium on Pulmonary Hypertension, in France, Pulmonary hypertension definition was revised by lowering the threshold from ≥25 mmHg to >20  It needs to be changed.
Reply: Thank you for the valuable suggestions. We have corrected the definition of pulmonary hypertension as the updated definition as below:
Pulmonary hypertension (PH) is defined as an elevation of mean pulmonary arterial pressure (mPAP) >20 mmHg associated with a pulmonary vascular resistance ≥3 Wood Units[1], which is divided into 5 major categories and more than 30 subcategories. (1. Simonneau G, Montani D, Celermajer DS, Denton CP, Gatzoulis MA, Krowka M, Williams PG, Souza R: Haemodynamic definitions and updated clinical classification of pulmonary hypertension. The European respiratory journal 2019, 53(1):1801913.)
- The major issue I have is that authors have chosen echo parameter for diagnosing pulmonary hypertension. In many patients RHC is needed to confirm pulmonary hypertension. Did authors confirmed diagnosis of PH with RHC? I saw that towards the end authors have mentioned it in the limitation which is a major limitation of the study.
Reply: We thank the reviewer for pointing out this issue. In this study, we retrospectively analyzed patients diagnosed with PH by echo parameter instead of RHC, and strictly abide by 2015 ESC/ERS guidelines, with a high level of probability of PH. In addition, we also referred to several retrospective studies that used the same or similar echocardiographic criteria to diagnose PH (Musumeci MB, et al. Pulmonary hypertension and clinical correlates in hypertrophic cardiomyopathy. Int J Cardiol. 2017;248:326-332; Bush D, et al. Clinical Characteristics and Risk Factors for Developing Pulmonary Hypertension in Children with Down Syndrome. J Pediatr. 2018;202:212-219.).
We edited the details about the study cohort regarding the comments and change as below:
We retrospectively analyzed 65 patients diagnosed with PH who underwent CMR scanning between February 2017 and March 2020. The inclusion criteria were those of established diagnostic criteria according to 2015 ESC/ERS guidelines[6] and patients diagnosed with PH by echocardiography were enrolled (Peak velocity of tricuspid regurgitation range 2.9-3.4 m/s, with presence of other echo “PH signs”; or Peak velocity of tricuspid regurgitation>3.4 m/s, without presence of other echo “PH signs”). Exclusion criteria: 1) Patients with any underlying cardiomyopathy (such as hypertrophic, amyloid, or ischemic cardiomyopathy); 2) Arrhythmia; 3) Massive pericardial effusion; 4) Substandard image quality; 5) Contraindication for an MR examination. For controls, 32 healthy subjects without any cardiovascular disease symptoms and with normal electrocardiogram (ECG) results were included.
- It was good to know that strain patterns were different between patients with PH compared to healthy subjects. Can authors also compare strain results among PH patients with preserved RV function vs declined RV function? I will be interested in knowing these findings.
Reply: Thank you for the valuable suggestions. We edited table 4 as the reviewer’s sugguestion as below, we also revised corresponding Statistical analysis, Results and Discussion.
Table 4 Comparison of RV strain parameters among PH patients with RV function reduction, preservation and healthy controls.
|
|
RVEF<40% group (n=51) |
RVEF≥40% group (n=14) |
Healthy control group (n=32) |
P |
|
RVGRS% |
10.68±5.12ab |
16.79±5.31b |
23.33±7.98 |
<0.001 |
|
RVRLS% |
-10.34±5.43ab |
-13.36±4.23b |
-20.08±6.41 |
<0.001 |
|
RVGCS% |
-6.80±3.07ab |
-10.28±3.00b |
-13.30±3.61 |
<0.001 |
|
RVGCS-b% |
-6.90±3.79 |
-6.70±6.08 |
-7.45±8.52 |
0.896 |
|
RVGCS-m% |
-6.96±4.45ab |
-11.84±3.16b |
-15.36±3.06 |
<0.001 |
|
RVGCS-a% |
-4.32±1.28ab |
-11.88±7.675b |
-19.08±5.49 |
<0.001 |
|
RVGRS-b% |
11.40±4.31b |
12.79±8.18b |
18.48±7.53 |
<0.001 |
|
RVGRS-m% |
11.33±6.50ab |
18.62±5.87b |
27.99±11.90 |
<0.001 |
|
RVGRS-a% |
11.58±4.30ab |
22.87±12.88b |
38.07±18.69 |
<0.001 |
Note: RV, right ventricle; global radial strain, GRS; global circumferential strain, GCS; global longitudinal strain, GLS; GRS for apical, GRS-a; GRS for mid segment, GRS-m; GRS for basal, GRS-b; GCS for apical GCS-a; GCS for mid segment, GCS-m; GCS for basal, GCS-b.
aP < 0.01 vs. RVEF≥40% group.
bP < 0.01 vs. controls.
Statistical analysis
SPSS version 22.0 software (SPSS, Chicago, IL) was used to conduct the statistical analyses. Continuous data were expressed as mean±standard deviation (SD).. The Shapiro-Wilk test was used to test the normality of the strain parameters, and two independent samples t-tests or nonparametric rank-sum tests were selected to compare the differences between two groups. We used single-factor ANOVA analysis for HF patients with reduced ejection fraction, HF patients with preserved ejection fraction, and healthy controls. The relationship between LV and RV strain parameters in the PH group were assessed using Pearson’s correlation analysis. P <0.05 indicated a statistically significant difference
Other modifications are shown in the revised manuscript.
Furthermore, we edited table 5 to added LVEF<60% group information.
Table 5 Comparison of LV strain parameters between PH patients with LV function reduction, preservation and healthy controls
|
|
LVEF<60% group (n=26) |
LVEF≥60% group (n=39) |
Healthy control group (N=32) |
P |
|
LVGRS% |
21.94±8.41ab |
36.20±7.90 |
36.31±7.92 |
<0.001 |
|
LVRLS% |
-10.30±3.27ab |
-14.16±3.04b |
-16.79±2.86 |
<0.001 |
|
LVGCS% |
-13.87±4.21ab |
-19.57±2.84 |
-19.91±2.90 |
<0.001 |
|
LVGCS-b% |
-13.68±3.94ab |
-18.31±2.75 |
-18.81±2.75 |
<0.001 |
|
LVGCS-m% |
-13.35±4.38ab |
-19.06±3.36 |
-19.80±3.06 |
<0.001 |
|
LVGCS-a% |
-15.31±5.13ab |
-24.50±2.96 |
-24.43±4.29 |
<0.001 |
|
LVGRS-b% |
19.83±10.35ab |
32.39±7.23 |
33.51±7.31 |
<0.001 |
|
LVGRS-m% |
20.51±8.34ab |
34.04±9.19 |
35.34±9.04 |
<0.001 |
|
LVGRS-a% |
30.40±18.35ab |
55.59±14.57 |
53.64±18.10 |
<0.001 |
Note: LV, left ventricle; global radial strain, GRS; global circumferential strain, GCS; global longitudinal strain, GLS; GRS for apical GCS-a; GRS for mid segment, GRS-m; GRS for basal GCS-b; GCS for apical GCS-a; GCS for mid segment, GCS-m; GCS for basal GCS-b
aP < 0.01 vs. LVEF≥60% group.
bP < 0.01 vs. controls.
Minor comments:
- Please review the spelling and grammar thoroghly throughout the paper. For example. I think in abstract, result section: line number 22, authors probably wanted to say left "ventricle" and not left "eye".
Reply: Thank for your comments. We have thoroughly checked and corrected the grammatical errors and typos we found in our revised manuscript by a native English speaker. If additonal modifications are needed, we will make further refinements.
We have modified the sentences as suggested and marked red.
- Line number 189 "Among the 65 patients in the PH group, there were 39 patients with LV function 188 preservation (RVEF ≥ 60%)". Authors probably want to say LVEF >60% in parenthesis.
Reply: The authors have revised the sentence as below:
Among the 65 patients in the PH group, there were 39 patients with LV function preservation (LVEF ≥ 60%).
- Did authors look into whether worse GRS, GCS or GLS correlate with mortality? I understand this many not be the primary focus of study and may not have been collected.
Reply: Thank you for your comment. We did not include studies on the association of myocardial strain and mortality, cause no outcome events have yet occurred during our follow-up. We would be very interested if outcome events appear in subsequent studies.
7.Authors should highlight that it is a retrospective study and is subject to limitations of retrospective study and genralization of these results should be made with caution.
Reply: We have restructured Discussion in the revised manuscript.

Reviewer 2 Report
Cao J. and colleagues conducted a retrospective study to analyze the LV and RV strain parameters in pulmonary hypertension (PH), including 65 PH patients and 32 healthy controls.
The study suffers from many methodological issues limiting its publication on the Journal of Clinical Medicine.
- It’s not clear what is the purpose of the paper.
- The definition of PH is not updated to the last World Symposium on Pulmonary Hypertension (6th WSPH) proposing to lower the mean pulmonary arterial pressure threshold from 25 mmHg to 20 mmHg combined to a pulmonary vascular resistance > 3 Wood Units (Simonneau G, Montani D, Celermajer DS, Denton CP, Gatzoulis MA, Krowka M, Williams PG, Souza R. Haemodynamic definitions and updated clinical classification of pulmonary hypertension. Eur Respir J. 2019 Jan 24;53(1):1801913. doi: 10.1183/13993003.01913-2018).
- Study population is heterogeneous, mixing PH patients regardless of the etiology of pulmonary pressures elevation. It’s not reported the distribution of patients into the five groups according to the comprehensive clinical classification of PH.
- Patients enrolled in the study were diagnosed with PH by echocardiography. This is a strong limitation, leading to a certain rate of false positives and questioning the study findings. Enrolling patients with peak velocity of tricuspid regurgitation > 2.8 m/s and two or more other signs of PH, the analysis includes also patients with intermediate echocardiographic probability of PH.
- Exclusion criteria should be better defined. Severe arrhythmia and severe cardiovascular disease are unclear sentences.
- Were patients enrolled retrospectively or prospectively? Discordant information are reported in the Abstract and in Material and methods description.
- The Authors stated that invasive right heart catheterization (RHC) has poor repeatability. Please add a reference supporting this sentence and cite the study by Melillo C.A. et al. “Repeatability of Pulmonary Pressure Measurements in Patients with Pulmonary Hypertension” (https://doi.org/10.1513/AnnalsATS.202002-182RL) addressing this issue.
- The authors stated the RHC provides limited information about RV function, giving a misleading message to readers. Indeed, during RHC the pressure-volume loops provide indexes of RV function that are independent of loading conditions and thus reflect intrinsic myocardial function. However, given the role of RV function as a major determinant of clinical outcome and the invasive nature of RHC, in clinical practice physicians need non-invasive imaging methods for its assessment. The Authors should better express this concept.
- The Authors concluded that their findings provide a proof of concept for the role of quantitative tissue techniques in noncontrast methods for recognizing and monitoring patients with PH. However, these conclusions are not adequately supported by the reported study.
Moreover, the paper lacks the correlation of RV and LV strain parameters with clinical outcomes. The Authors should better discuss the contribution that RV and LV strain analysis could make in clinical practice for PH specialists.
- In the paper interclass correlation coefficients (ICCs) were designed to assess interobserver consistency. Indeed, preferably the Bland-Altman method should be used to evaluate the interobserver concordance.
- Pearson correlation analyses rather than Spearman should be used to determine the correlation of LV and RV strain parameters in the PH group.
- References are not numbered in order of appearance in the text and listed individually at the end of the manuscript, as reported in the instructions of the Journal of Clinical Medicine. Indeed, in the text, reference numbers should be placed in square brackets [ ], and placed before the punctuation. Moreover, all references should be described as follows: for Journal Articles 1. Author 1, A.B.; Author 2, C.D. Title of the article. Abbreviated Journal Name Year, Volume, page range. Please check the complete Instructions for Authors on the website.
Author Response
Dear Editors and Reviewers,
We would like to thank you for your careful reading, helpful comments, and constructive suggestions, which has significantly improved the presentation of our manuscript. It's an honor to have chance to publish our work in Journal of Clinical Medicine. We have carefully considered all comments from the reviewers and revised our manuscript accordingly. The manuscript has also been double-checked, and the typos and grammar errors we found have been corrected.
The followings are our RESPONSE to DEPUTY EDITOR & REVIEWER COMMENTS.
Cao J. and colleagues conducted a retrospective study to analyze the LV and RV strain parameters in pulmonary hypertension (PH), including 65 PH patients and 32 healthy controls.
The study suffers from many methodological issues limiting its publication on the Journal of Clinical Medicine.
- It’s not clear what is the purpose of the paper.
Reply: We are sorry for the confusing description. We have revised the objective as reviewer’s suggestion as below:
Objectives: To evaluate both right and left ventricular (LV and RV) function in patients with pulmonary hypertension (PH) by using cardiac magnetic resonance tissue tracking technology (CMR-TT), and to explore its clinical application value.
- The definition of PH is not updated to the last World Symposium on Pulmonary Hypertension (6th WSPH) proposing to lower the mean pulmonary arterial pressure threshold from 25 mmHg to 20 mmHg combined to a pulmonary vascular resistance > 3 Wood Units (Simonneau G, Montani D, Celermajer DS, Denton CP, Gatzoulis MA, Krowka M, Williams PG, Souza R. {Simonneau, 2019 #61}. Eur Respir J. 2019 Jan 24;53(1):1801913. doi: 10.1183/13993003.01913-2018).
Reply: We have corrected the definition of pulmonary hypertension as the updated definition as below:
Pulmonary hypertension (PH) is defined as an elevation of mean pulmonary arterial pressure (mPAP) >20 mmHg associated with a pulmonary vascular resistance ≥3 Wood Units[1], which is divided into 5 major categories and more than 30 subcategories. (1.Simonneau G, Montani D, Celermajer DS, Denton CP, Gatzoulis MA, Krowka M, Williams PG, Souza R: Haemodynamic definitions and updated clinical classification of pulmonary hypertension. The European respiratory journal 2019, 53(1):1801913.)
- Study population is heterogeneous, mixing PH patients regardless of the etiology of pulmonary pressures elevation. It’s not reported the distribution of patients into the five groups according to the comprehensive clinical classification of PH.
Reply: Indeed, it will be more convincing if we divided PH patients into the five groups. However, this study included retrospective identification of adult patients with PH who underwent clinical CMR scanning. The number of PH patients who underwent clinical CMR during the study was restricted, and as such, the etiologies of PH included in our study were heterogeneous. Patients with all WHO groups of PH were included in the study.
We have revised the limitation as reviewer’s suggestion.
First, the patients enrolled in this study were diagnosed with PH by echocardiography, and the literature suggests that although the echocardiographic tricuspid regurgitation method correlates well with the cardiac catheterization results, it is not a substitute technique[28], so there may be a certain rate of false-positives among the PH patients. Second, This study is a retrospective analyse of patients with PH who underwent clinical CMR scanning. The number of PH patients who underwent clinical CMR during the study was restricted, and as such, the etiologies of PH included in our study were heterogeneous. Third, there were differences in the sex ratio between the control and patient groups. Previous studies have shown sex differences in LV and RV strain in healthy individuals[29, 30], so the differences in the strain parameters between the two groups may be influenced by sex factors. Last, only the strain parameters were studied, and the strain rates of PH patients should be further analysed.
- Patients enrolled in the study were diagnosed with PH by echocardiography. This is a strong limitation, leading to a certain rate of false positives and questioning the study findings. Enrolling patients with peak velocity of tricuspid regurgitation > 2.8 m/s and two or more other signs of PH, the analysis includes also patients with intermediate echocardiographic probability of PH.
Reply: We thank the reviewer for pointing out this issue. In this study, we retrospectively analyzed patients diagnosed with PH by echo parameter instead of RHC, and strictly abide by 2015 ESC/ERS guidelines, with a high level of probability of PH. In addition, we also referred to several retrospective studies that used the same or similar echocardiographic criteria to diagnose PH (Musumeci MB, et al. Pulmonary hypertension and clinical correlates in hypertrophic cardiomyopathy. Int J Cardiol. 2017;248:326-332; Bush D, et al. Clinical Characteristics and Risk Factors for Developing Pulmonary Hypertension in Children with Down Syndrome. J Pediatr. 2018;202:212-219.).
We are sorry for the confusing description for “Enrolling patients with peak velocity of tricuspid regurgitation > 2.8 m/s and two or more other signs of PH”
We edited the details about the study cohort regarding the comments and change as below:
We retrospectively analyzed 65 patients diagnosed with PH who underwent CMR scanning between February 2017 and March 2020. The inclusion criteria were those of established diagnostic criteria according to 2015 ESC/ERS guidelines[6] and patients diagnosed with PH by echocardiography were enrolled (Peak velocity of tricuspid regurgitation range 2.9-3.4 m/s, with presence of other echo “PH signs”; or Peak velocity of tricuspid regurgitation>3.4 m/s, without presence of other echo “PH signs”). Exclusion criteria: 1) Patients with any underlying cardiomyopathy (such as hypertrophic, amyloid, or ischemic cardiomyopathy); 2) Arrhythmia; 3) Massive pericardial effusion; 4) Substandard image quality; 5) Contraindication for an MR examination. For controls, 32 healthy subjects without any cardiovascular disease symptoms and with normal electrocardiogram (ECG) results were included.
- Exclusion criteria should be better defined. Severe arrhythmia and severe cardiovascular disease are unclear sentences.
Reply: We have modified the exclusion criteria as suggested and marked red.
Exclusion criteria: 1) Patients with any underlying cardiomyopathy (such as hypertrophic, amyloid, or ischemic cardiomyopathy); 2) Arrhythmia; 3) Massive pericardial effusion; 4) Substandard image quality; 5) Contraindication for an MR examination.
- Were patients enrolled retrospectively or prospectively? Discordant information are reported in the Abstract and in Material and methods description.
Reply: All patients were enrolled retrospectively. We have restructured Abstract and Material and methods in the revised manuscript.
- The Authors stated that invasive right heart catheterization (RHC) has poor repeatability. Please add a reference supporting this sentence and cite the study by Melillo C.A. et al. “Repeatability of Pulmonary Pressure Measurements in Patients with Pulmonary Hypertension” (https://doi.org/10.1513/AnnalsATS.202002-182RL) addressing this issue.
Reply: Thank you for suggestions.We have revised Introduction Para.2 as below:
It has been reported that RV function is one of the major predictors of the outcome regardless of the aetiology of patients with PH[3-5]. The gold standard for diagnosing PH is right heart catheterization (RHC)[6]. Although RHC can evaluate pulmonary haemodynamics and has good repeatability[7]. In consideration of RV function is the main determinant of clinical outcome and the invasiveness of RHC, we need non-invasive imaging methods to evaluate it in clinical practice. Most studies have focused on the RV function. It is well known that increased pressure in the RV can lead to changes in interventricular septum morphology[8], and the morphology and function of the LV and RV are also mutually affected. Therefore, noninvasive evaluation of LV and RV function is vital.
- The authors stated the RHC provides limited information about RV function, giving a misleading message to readers. Indeed, during RHC the pressure-volume loops provide indexes of RV function that are independent of loading conditions and thus reflect intrinsic myocardial function. However, given the role of RV function as a major determinant of clinical outcome and the invasive nature of RHC, in clinical practice physicians need non-invasive imaging methods for its assessment. The Authors should better express this concept.
Reply: We have revised Introduction Para.2 as below:
It has been reported that RV function is one of the major predictors of the outcome regardless of the aetiology of patients with PH[3-5]. The gold standard for diagnosing PH is right heart catheterization (RHC)[6]. Although RHC can evaluate pulmonary haemodynamics and has good repeatability[7]. In consideration of RV function is the main determinant of clinical outcome and the invasiveness of RHC, we need non-invasive imaging methods to evaluate it in clinical practice. Most studies have focused on the RV function. It is well known that increased pressure in the RV can lead to changes in interventricular septum morphology[8], and the morphology and function of the LV and RV are also mutually affected. Therefore, noninvasive evaluation of LV and RV function is vital.
- The Authors concluded that their findings provide a proof of concept for the role of quantitative tissue techniques in noncontrast methods for recognizing and monitoring patients with PH. However, these conclusions are not adequately supported by the reported study.
Moreover, the paper lacks the correlation of RV and LV strain parameters with clinical outcomes. The Authors should better discuss the contribution that RV and LV strain analysis could make in clinical practice for PH specialists.
Reply: Thank you for the valuable suggestions. We have restructured Discussion in the revised manuscript.
- In the paper interclass correlation coefficients (ICCs) were designed to assess interobserver consistency. Indeed, preferably the Bland-Altman method should be used to evaluate the interobserver concordance.
Reply: The Repeatability test and Reproducibility analysis were revised based on reviewer’s suggestion.
Twenty patients with PH were randomly selected, and CMR-TT postprocessing was performed by two radiologists. They were blinded to the information of the patients. Consistency was tested by Bland-Altman analysis to calculate the 95% limits of mean deviation and consistency.
The radiologists were in good agreement for their interpretations. The results showed preferable repeatability of CMR-TT technology in analysing the global radial strain (GRS), circumferential strain (GCS) and longitudinal strain (GLS) of LV and RV in the PH group (Table 2) and the LVGCS had the best inter-observer agreement (95% CI, 0.956-0.994).
- Pearson correlation analyses rather than Spearman should be used to determine the correlation of LV and RV strain parameters in the PH group.
Reply: We are very sorry for the mistakes in the statistical analysis. Yes, we agree with the review, we have conducted the Pearson correlation analyses to determine the correlation. The authors have revised the Statistical analysis as below:
SPSS version 22.0 software (SPSS, Chicago, IL) was used to conduct the statistical analyses. Measurement data are expressed as the mean±SD. The Shapiro-Wilk test was used to test the normality of the strain parameters, and two independent samples t-tests or nonparametric rank-sum tests were selected to compare the differences between two groups. The relationship between LV and RV strain parameters in the PH group were assessed using Pearson’s correlation analysis. P <0.05 indicated a statistically significant difference.
- References are not numbered in order of appearance in the text and listed individually at the end of the manuscript, as reported in the instructions of the Journal of Clinical Medicine. Indeed, in the text, reference numbers should be placed in square brackets [ ], and placed before the punctuation. Moreover, all references should be described as follows: for Journal Articles 1. Author 1, A.B.; Author 2, C.D. Title of the article. Abbreviated Journal Name Year, Volume, page range. Please check the complete Instructions for Authors on the website.
Reply: As suggested by the reviewer, we have revised the format of references.

Round 2
Reviewer 1 Report
Thank you for addressing the comments. The Paper looks good now.
Author Response
Thank you for your help, your comments have made the manuscript improve a lot.
Reviewer 2 Report
The study enrolled PH patients diagnosed by echocardiography: peak velocity of tricuspid regurgitation range 2.9-3.4 m/s, with presence of other echo PH signs or peak velocity of tricuspid regurgitation > 3.4 m/s without presence of other echo PH signs. Therefore, population included in the study is heterogeneous, mixing different groups of PH according to the comprehensive clinical classification of PH and including a certain rate of false positives.
Indeed, as reported in the 2015 ESC/ERS guidelines, echocardiography helps clinicians in grading the probability of PH but its not sufficient for diagnosis of PAH (group 1) and CTEPH (group 4).
Authors’ reply is questionable. The study by Musumeci et al. included patients with hypertrophic cardiomyopathy (HCM) who can develop PH due to elevated left-sided diastolic pressure (PH group 2). In these patients right heart catheterization is not mandatory in the diagnostic algorithm but is recommended only in specific situations.
Moreover, patients with any underlying cardiomyopathy such as HCM were excluded from the study by Cao J. and colleagues.
Finally, the paper still has a poor English level.
Author Response
Dear Editors and Reviewers,
We would like to thank you for your careful reading, helpful comments, and constructive suggestions, which has significantly improved the presentation of our manuscript. It's an honor to have chance to publish our work in Journal of Clinical Medicine. We have carefully considered all comments from the reviewers and revised our manuscript accordingly. The manuscript has also been double-checked, and the typos and grammar errors we found have been corrected.
The followings are our RESPONSE to DEPUTY EDITOR & REVIEWER COMMENTS.
Comments and Suggestions for Authors
The study enrolled PH patients diagnosed by echocardiography: peak velocity of tricuspid regurgitation range 2.9-3.4 m/s, with presence of other echo PH signs or peak velocity of tricuspid regurgitation > 3.4 m/s without presence of other echo PH signs. Therefore, population included in the study is heterogeneous, mixing different groups of PH according to the comprehensive clinical classification of PH and including a certain rate of false positives.
Indeed, as reported in the 2015 ESC/ERS guidelines, echocardiography helps clinicians in grading the probability of PH but its not sufficient for diagnosis of PAH (group 1) and CTEPH (group 4).
Authors’ reply is questionable. The study by Musumeci et al. included patients with hypertrophic cardiomyopathy (HCM) who can develop PH due to elevated left-sided diastolic pressure (PH group 2). In these patients right heart catheterization is not mandatory in the diagnostic algorithm but is recommended only in specific situations.
Moreover, patients with any underlying cardiomyopathy such as HCM were excluded from the study by Cao J. and colleagues.
Finally, the paper still has a poor English level.
Thank you for your helpful comments, and constructive suggestions, which has significantly improved the presentation of our manuscript. We have carefully considered all comments from the reviewers and revised our manuscript accordingly. The manuscript has also been double-checked, and the typos and grammar errors we found have been corrected.
We have revised the data based on RHC as a gold standard, and only PAH patients (group 1) patients were included.We have edited the details about the study cohort as below:
We retrospectively analysed 47 patients diagnosed with PAH (World Health Organization (WHO) Group 1) who underwent CMRI between February 2017 and March 2020. The inclusion criteria were those of established according to the 2015 ESC/ERS guidelines [8]. All patients enrolled in this study were diagnosed with PAH by echocardiography (ECG), history taking, and RHC. Thirty-nine PAH subjects had idiopathic PAH, six had PAH associated with systemic lupus erythematosus (SLE), and two had PAH associated with systemic sclerosis. In our cohort, PAH was defined as mPAP ≥ 25 mmHg at rest and PAWP was ≤ 15 mmHg obtained from RHC. The exclusion criteria were as follows: 1) any underlying cardiomyopathy (such as hypertrophic, amyloid, or ischaemic cardiomyopathy); 2) arrhythmia; 3) lung disease/hypoxia, chronic thromboembolism; 4) massive pericardial effusion; 5) substandard image quality; and 6) contraindication to MRI. As controls, 32 healthy subjects without any cardiovascular disease symptoms and with normal ECG results were included.
The language presentation have been improved with assistance from a native English speaker with appropriate research background.
